# *In vitro–in silico* correlation of three-dimensional turbulent flows in an idealized mouth-throat model

Eliram Nof[1¤a], Saurabh Bhardwaj[1¤b], Pantelis Koullapis[2], Ron Bessler[1], Stavros Kassinos[2], Josué Sznitman[1] *

**1** Department of Biomedical Engineering, Technion—Israel Institute of Technology, Haifa, Israel,
**2** Computational Sciences Laboratory (UCY-CompSci), Department of Mechanical and Manufacturing Engineering, University of Cyprus, Nicosia, Cyprus

¤a Current address: Department of Medical Physics, Memorial Sloan Kettering Cancer Center, New York, New York, United States of America
¤b Current address: Department of Biomedical Engineering, Pennsylvania State University, State College, Pennsylvania, United States of America
* sznitman@bm.technion.ac.il

**Data Availability Statement:** Relevant data are within the manuscript and its Supporting information files, and in addition we have provided

## Abstract

There exists an ongoing need to improve the validity and accuracy of computational fluid dynamics (CFD) simulations of turbulent airflows in the extra-thoracic and upper airways. Yet, a knowledge gap remains in providing experimentally-resolved 3D flow benchmarks with sufficient data density and completeness for useful comparison with widely-employed numerical schemes. Motivated by such shortcomings, the present work details to the best of our knowledge the first attempt to deliver *in vitro–in silico* correlations of 3D respiratory air-flows in a generalized mouth-throat model and thereby assess the performance of Large Eddy Simulations (LES) and Reynolds-Averaged Numerical Simulations (RANS). Numerical predictions are compared against 3D volumetric flow measurements using Tomographic Particle Image Velocimetry (TPIV) at three steady inhalation flowrates varying from shallow to deep inhalation conditions. We find that a RANS k-$\omega$ SST model adequately predicts velocity flow patterns for Reynolds numbers spanning 1'500 to 7'000, supporting results in close proximity to a more computationally-expensive LES model. Yet, RANS significantly underestimates turbulent kinetic energy (TKE), thus underlining the advantages of LES as a higher-order turbulence modeling scheme. In an effort to bridge future endevours across respiratory research disciplines, we provide end users with the present *in vitro–in silico* correlation data for improved predictive CFD models towards inhalation therapy and therapeutic or toxic dosimetry endpoints.

## Author summary

The dispersion and ensuing deposition of inhaled airborne particulate matter in the lungs are strongly influenced by the dynamics of turbulent respiratory airflows in the mouth-throat region during inhalation. To cirumvent costly *in vitro* experimental measurement

a link to a repository: https://doi.org/10.6084/m9.figshare.20239125.

**Funding:** This work was supported by the Israel Science Foundation (ISF) (grant no. 1840/21 to JS), the European Research Council (ERC) under the European Union's Horizon 2020 research and innovation program (grant agreement no. 677772 to JS) as well as the PBC Fellowship Program by the Israeli council for supporting SB with partial funding. The funders had no role in study design, data collection and analysis, decision to publish, or preparation of the manuscript.

**Competing interests:** N/A.

resources, fluid dynamics (CFD) simulations are widely sought to predict deposition outcomes but often lack detailed experimental data to first validate the three-dimensional (3D) flow structures anticipated to arise in the upper respiratory tract. In an effort to reconcile such data scarcity, we deliver experimental-numerical correlations of 3D respiratory airflows in an idealized 3D printed mouth-throat model against two widely-established numerical schemes with varying computational costs, namely coarse RANS and finer LES technique. Our time-resolved 3D flow data underline the complexity of these physiological inhalation flows, and discuss advantages and drawbacks of the different numerical techniques. With an outlook on future respiratory applications geared towards broad preclinical inhaled aerosol deposition studies, our open source data are made available for future benchmark comparisons for a broad range of end users in the respiratory research community.

## Introduction

Respiratory airflow characteristics are known to strongly influence the transport and deposition of inhaled aerosols in the human airways. The intricate geometry of the extra-thoracic that includes bends, expansions and constrictions leads to transition to turbulent airflows in the pharynx, larynx and trachea with Reynolds numbers on the order of several thousands (i.e., 2'000–10'000), depending on inhalation regimes [1–4]. Detailed modeling of these complex flows is required to determine the fate of particle-laden airflows and ensuing deposition patterns towards predicting for example pulmonary dosimetry [5] or the dispersion of airborne pathogen in the lungs [6, 7].

In recent years, computational fluid dynamics (CFD) have spearheaded advances to overcome some of the prohibitive costs of *in vitro* experimental campaigns. Numerical methods solving the Reynolds-Averaged Navier-Stokes (RANS) equations are the most widely adopted for modeling non-laminar flows due to their lower computational cost. RANS typically involves applying a turbulence model to a stationary solution at steady breathing conditions [8]. Coupled with the low-Reynolds number (LRN) k–$\omega$ turbulence model (e.g. k–$\omega$ SST), RANS is often used to predict laminar–transitional–turbulent flows in the respiratory tract [9, 10]. While RANS simulations are popular, they often come short of determining model constants, making clinical relevance and comparison with other studies challenging [11–15]. Riazuddin et al. [16] used a k-$\omega$ SST turbulence model to investigate breathing in a nasal cavity, demonstrating the model's accuracy for modeling flows with unfavourable pressure gradients via good agreement with experimental and numerical data. Ma et al. simulated airflows and aerosol transport in patient-derived human airways with a k-$\epsilon$ model, showing good agreement with coarse (i.e., regionally averaged) *in vivo* deposition data [17]. Stapleton et al. similarly studied aerosol deposition using a k-$\epsilon$ model in an idealized mouth–throat validated with *in vitro* regional deposition measurements from gamma scintigraphy [18]. The authors found good agreement for laminar but not turbulent conditions, suggesting that particle deposition may be sensitive to pressure drop and flow recirculation. Longest et al. investigated local airway aerosol deposition using different variants of the k-$\omega$ turbulence model [19], finding best agreement with *in vitro* deposition patterns using the low Reynolds number (LRN) approximation. Despite such popularity, most RANS studies have used (sub-)regionally averaged deposition metrics for experimental validation, whereas more accurate and spatially-resolved deposition models would first require experimental validation of the underlying turbulent flows.

As RANS modelling is not sufficiently accurate to capture small-scale fluctuations in turbulent flows in extra-thoracic airways, a more adequate choice lies in Large Eddy Simulations (LES). There, only the smallest flow scales containing a small fraction of the kinetic energy are discarded, thereby retaining significantly more features of the underlying turbulence physics compared with RANS [20, 21]. Lin et al. [20] were among the first to use LES in assessing the effect of laryngeal jet-induced turbulence on airflow characteristics and tracheal wall shear stress. Their study revealed that turbulence generated by the laryngeal jet can significantly affect the downstream flow patterns, highlighting the importance of including the extrathoracic airways in a model. Choi et al. [2] performed LES in two CT-derived upper airway models to examine the effect of inter-subject variabilities on the overall flow characteristics, finding that the glottis constriction ratio and the curvature and shape of the airways have significant effect on the generated flows. More recently, Koullapis et al. used LES to investigate inlet flow conditions in a CT-reconstructed geometry of the human airways [21] where flow field differences largely dissipated just a short distance downstream of the mouth inlet. Furthermore, increasing the inhalation flowrate from sedentary to active breathing conditions left the mean flow field structures largely unaffected. Recent LES-based deposition studies have established the trustworthiness of the method using generalized geometries [22, 23] meanwhile constant gains in computing power have made LES more affordable. However, the underlying computational expense of LES is still considerably higher than RANS, precluding its accessibility for broader use. Furthermore, imaging modalities are generating larger and more intricate patient geometries, requiring first the validation of simpler models for practical and patient-specific applications.

To this end, a European Cooperation in Science and Technology (COST) action published a much needed benchmark case, known as the Siminhale benchmark [23], comparing several numerical schemes in an idealized airway geometry spanning mouth to the fourth bronchial generation at an inhalation flowrate of 60 l/min. *In silico* predictions were validated against 2D particle image velocimetry (PIV) measurements [24] with good agreement between PIV and LES, with a slight over-prediction of turbulent kinetic energies (TKE) in the simulations. The experimental and numerical data in [24], along with two additional LES and a RANS datasets, are currently part of a common publicly accessible ERCOFTAC database providing best practice advice for setting up a computational fluid-particle dynamics (CFPD) model of the human upper airways with available validation data. However, the PIV data are limited to six orthogonal 2D planes and the laryngeal constriction, known to strongly modulate the inlet flow [25], was not captured due to obstruction from optical access. Several studies have measured flow in the laryngeal site using planar PIV alone [26, 27], while others employed numerical methods without comparable experimental measurements. Jayaraju et al. validated their LES simulations in a mouth-throat model with planar PIV, but measured only at the mid-sagittal plane [22]. In turn, a knowledge gap still remains in providing an experimental benchmark with sufficient data density and completeness for useful comparison to numerical schemes.

Recently, volumetric flow measurement tools have been leveraged to study respiratory flows in 3D with tomographic particle image velocimetry (TPIV) and magnetic resonance velocimetry (MRV) [28, 29]. Notably, Kenjeres and Tijn validated their RANS and LES simulations with available 3D MRV data [28] in an identical upper airway geometry [30] but the use of a patient-specific model limits validations for future comparative studies. Motivated by these ongoing shortcomings, the present work details the first attempt to deliver *in vitro–in silico* flow correlations in a generalized mouth-throat model capturing 3D airflow patterns. RANS and LES predictions are compared against 3D TPIV flow measurements at three steady inhalation flowrates varying from shallow to deep inhalation conditions. We share our model

geometry and TPIV measurement data https://doi.org/10.6084/m9.figshare.20239125 as open-source material towards future benchmark references.

## Materials and methods

### Geometry and flow conditions

The oral airway is based on a simplified elliptic model extending from the mouth through the larynx and previously used in aerosol inhalation studies [31]. This elliptic mouth-throat (MT) model generalizes a patient-specific geometry derived from a healthy adult's computed tomography (CT) scan [32].

Comparisons between *in vitro* measurements and *in silico* simulations are carried out at three distinct Reynolds numbers, namely Re = 1'500, 4'500 and 7'000 based on the inlet mouth diameter of the model. Values of Re correspond to steady inhalation air flowrates of approximately 11.5, 32.8 and 52 l/min; such inhalation conditions are comparable to sedentary, light and heavy exercise conditions, respectively. Moreover, the two latter inhalation flowrates are specifically relevant for pulmonary drug delivery via a dry powder inhaler (DPI) [33].

### Experimental method

We briefly describe the experimental setup (Figs 1 and S1. A closed-loop perfusion system comprising a centrifugal pump, 15 l reservoir tank and digital flow rate sensor, supplies water/glycerol (58:42 mass ratio with a density of $\rho_f$ = 1'150 kg/m$^3$ and a dynamic viscosity of $\mu_f$ = 9.66 × 10$^{-3}$ kg/m-s) through the phantom model (fabricated using previously described methods [34, 35]). We measure the three-dimensional, three-component (3D-3C) velocity fields in the phantom model using a high-speed, tomographic particle image velocimetry setup (TPIV) (LaVision GmbH, Germany) consisting of four CMOS cameras (Fastcam Mini UX100, 1'280 x 1'024 pixel, 12 bit, Photron USA, Inc.) equipped with 100 mm focal length lenses (Zeiss Milvus, Germany). Volume illumination is provided by a 70 mJ dual-head Nd:YLF laser (DM30–527DH, Photonics Industries, USA) and continuous image acquisition was conducted using a frame straddling technique at a fixed 1'250 frames per second (fps), where the time separation between laser pulses was set to 25, 40 and 65 $\mu$s, respectively for the high, mid and low Re cases investigated. The field of view (FOV) spans 18.5 × 45 × 9 mm (*x-y-z*). These experimental settings were carefully optimized to ensure that individual seeded particles correspond to 3 to 5 pixels imaged in the instantaneous images so to avoid any peak-locking effects for PIV while maximum particle displacements betwee consecutive images range between 3 and 8 pixels depending on the flow case. The laser beam is introduced through the side of the model and shaped into a thick slab by an optical arrangement consisting of a beam expander and cylindrical lens, followed by a knife edge aperture (Fig 1a); the latter is commonly used in PIV to reduce light reflections from regions void of tracer particles, contributing amongst other to the rectangularity of the processed vector maps relative to the actual elliptical shape of the experimental model (see results).

Details on the TPIV methodology, including refractive index matching and scaling following dynamic similarity can be found in our previous work [35], where raw images and TPIV processing are performed with Davis 10 (LaVision GmbH, Germany) and further analyzed in Matlab (Mathworks Inc., USA). Briefly, red fluorescent polystyrene particles (PS-FluoRed, microParticles GmbH, Germany) are seeded and act as flow tracers where optical filters are fitted to each camera lens to reduce non-fluorescent light reflection thereby increasing signal-to-noise (SNR) ratio. The mean particle diameter $d_p$ = 10 $\mu$m and particle density $\rho_p$ = 1'050 kg/

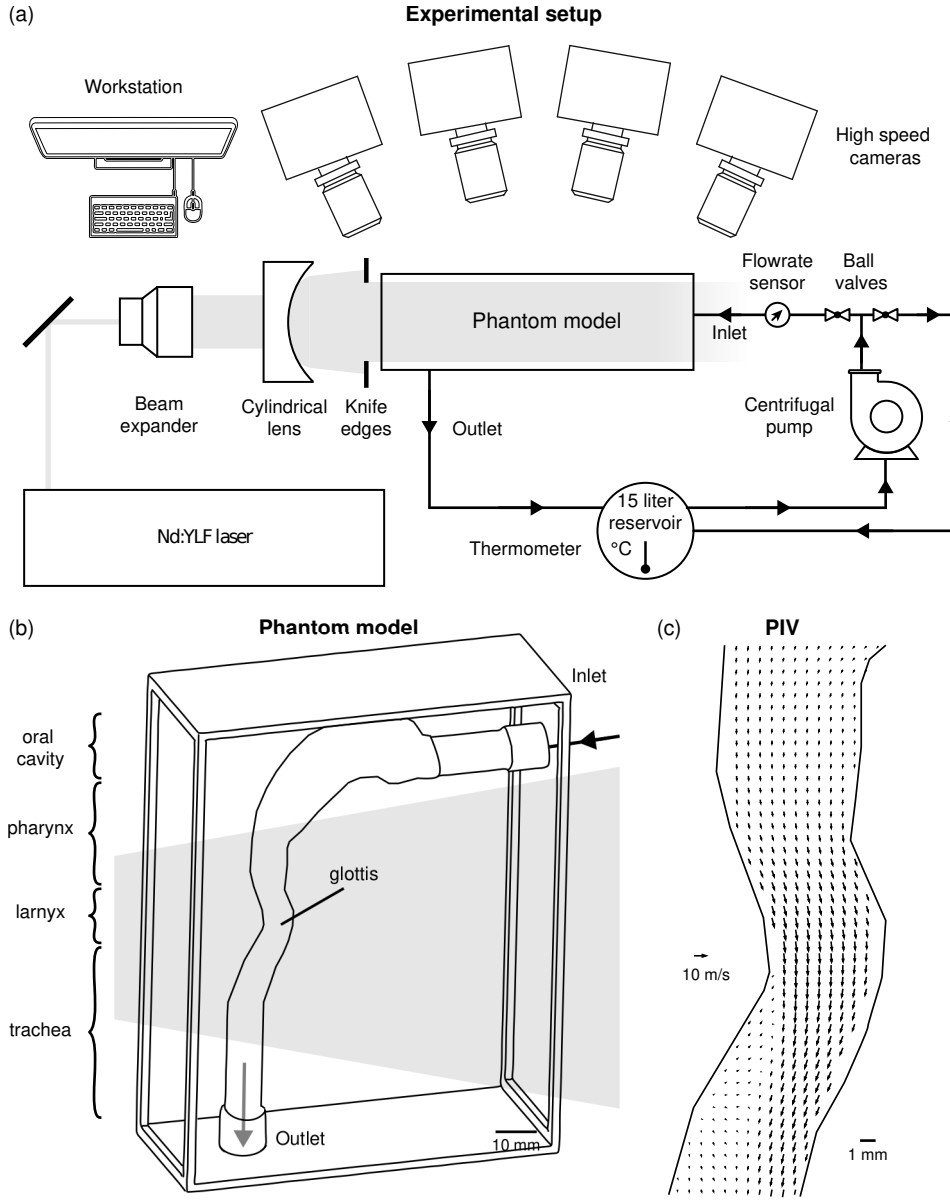

**Fig 1. Model geometry and tomographic particle image velocimetry (TPIV) setup.** (a) Experimental setup consisting of a laser, optical equipment, flow system, four high-speed cameras and a phantom idealized mouth-throat model. (b) The phantom model is illustrated schematically in three-dimensional view with inlet and outlet ports marked. Note that the laryngeal constriction is illuminated by laser light, maximizing spatial resolution in the specific region of interest (ROI). (c) A representative vector field is plotted along the mid-sagittal plane, following TPIV processing algorithms.

m³ yield a corresponding particle relaxation time $\tau_s = d_p^2\rho_p/18\mu_f$ of 0.6 $\mu$s and equivalent to a particle Stokes number much smaller than unity. Concurrently, particle drift due to buoyancy effects resulting from density differences between the working fluid and the particle, are largely negligible as the (buoyant) terminal velocity is estimated to be $u_t = 5.7 \times 10^{-7}$ m/s (i.e. $u_t = gd_p^2(\rho_f - \rho_p)/18\mu_f$), compared with characteristic velocities of the flow on the order of $\mathcal{O}(1)$ m/s at the mouth inlet.

## Numerical methods

For the CFD simulations, we introduce the governing equations and computational methodology used to describe the motion of air based on the Eulerian approach. The governing equations for incompressible fluid flow are composed of the Navier-Stokes' (momentum) and continuity equations. Two distinct numerical methodologies (RANS and LES) are adopted to model turbulent flows in the mouth-throat geometry.

**RANS details.** A commercial software (ANSYS Fluent, ANSYS Inc.) was used to perform the transient flow simulations using a RANS approach in which mass and momentum (i.e. Navier-Stokes) conservation equations are solved numerically by using the finite volume method (FVM) in the 3D domain (see S1 Text. for RANS equations). Turbulent flow phenomena were modeled using the Shear Stress Transport (SST) k-$\omega$ model with Low Reynolds Number (LRN) correction; considered the most suitable RANS model for predicting low turbulence flow in the respiratory system, in particular with limited available computational resources. The popular SST k-$\omega$ turbulence model [36, 37] is a two-equation eddy-viscosity model. The use of a k-$\omega$ formulation in the inner parts of the boundary layer renders the model directly usable all the way down to the wall through the viscous sub-layer. Hence the SST k-$\omega$ model can be used as a Low-Re turbulence model without any extra damping functions. The SST formulation also switches to a k-$\epsilon$ behaviour in the free-stream and thus avoids the common k-$\omega$ problem whereby the model is too sensitive to the inlet free-stream turbulence properties. Furthermore, the SST k-$\omega$ model is often credited for its good behaviour in adverse pressure gradients and separated flows [3, 38]. Note that the SST k-$\omega$ model is acknowledged to produce slightly high turbulence levels in regions with large normal strains, e.g. stagnation regions and regions with strong acceleration. However, this tendency is much less pronounced than with a normal k-$\epsilon$ model [39, 40].

The computational geometry was discretized in ICEM (ANSYS Inc., Canonsburg, PA) using tetrahedral elements with prism layers at the walls. The final mesh was transformed into a polyhedral mesh in Fluent (ANSYS, Inc.). A rigorous mesh convergence study was first carried out (i.e. ranging from 2M to 6M tetrahedral cells) to eventually select the final mesh of $\sim$900'000 polyhedral cells (converted from $\sim$2.4M tetrahedral cells), with up to 10 prism layers for near-wall refinements (see S2 Fig). For each Re condition, a fully-developed parabolic velocity profile is provided at the inlet surface whereas outflow condition is prescribed at outlet surface and guaranteeing no-slip velocity conditions at the walls. A second-order implicit scheme is used for the transient formulation, with a time step of $10^{-3}$ s to ensure good accuracy. To decrease numerical diffusion in the unstructured three-dimensional mesh, a second-order upwind approach is utilized to discretize the advection terms. A segregated solver is then used to solve the resultant system of equations. The SIMPLE algorithm is used to solve the governing equations by coupling velocity and pressure.

**LES details.** Large Eddy Simulations (LES) are performed using the dynamic version of the Smagorinsky-Lilly subgrid scale model [41] in order to examine the unsteady flow in the upper airways geometry. Previous studies have shown that this model performs well in transitional flows in the human airways [24]. The airflow is described by the filtered set of incompressible Navier-Stokes equations.

In order to generate appropriate inlet velocity conditions for the CFD model, a mapped inlet (or recycling) boundary condition is used [42]. To apply this boundary condition, the pipe at the inlet is extended by a length equal to ten times its diameter. The pipe section is initially fed with an instantaneous turbulent velocity field generated in a separate pipe flow LES. During the simulation, the velocity field from the mid-plane of the pipe domain is mapped to the inlet boundary. Scaling of the velocities is applied to enforce the specified bulk flow rate. In

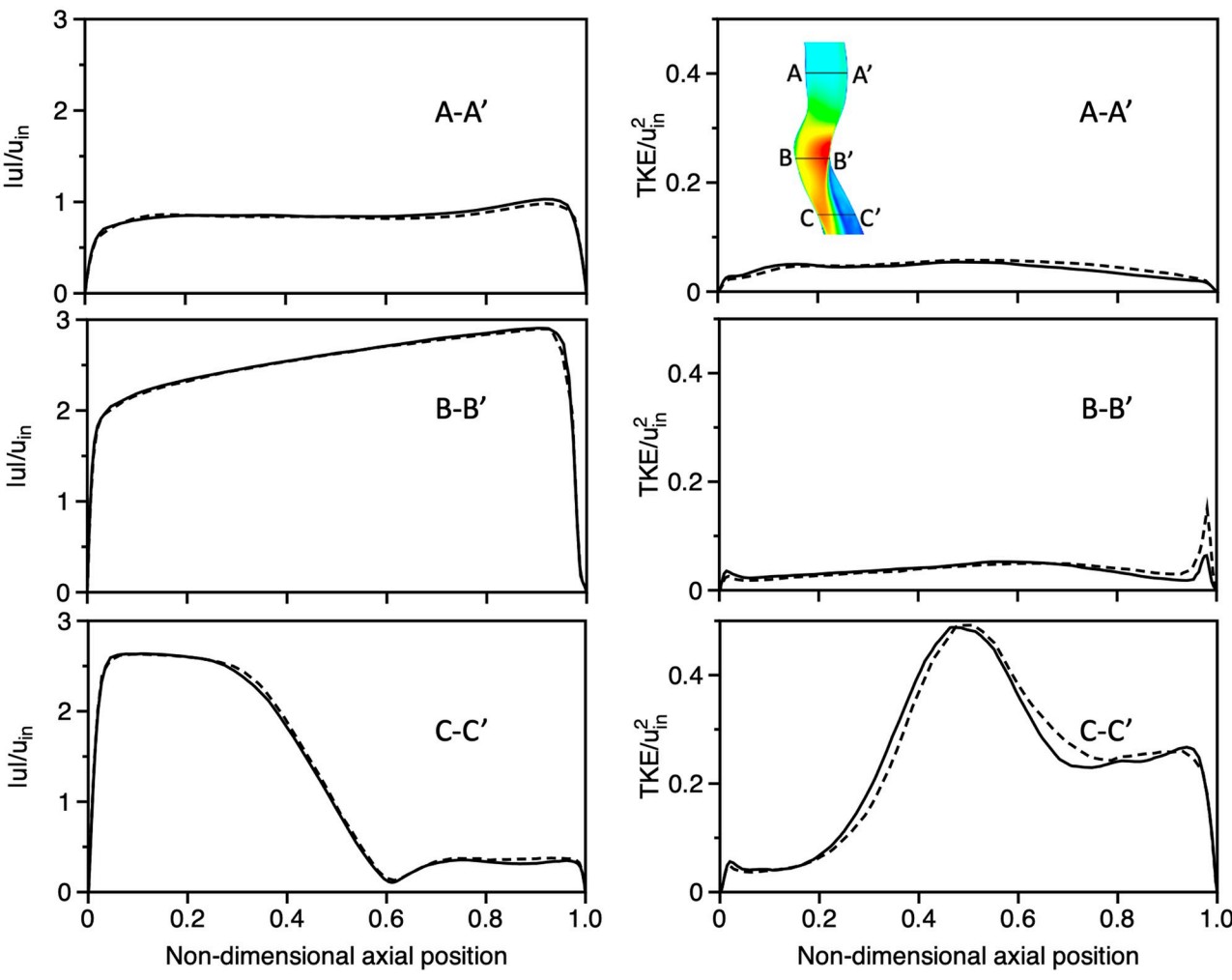

**Fig 2. Large Eddy Simulation (LES) grid sensitivity analysis (Re = 7'000).** Comparison of 1D profiles of normalized time-averaged velocity magnitudes (first column) and turbulent kinetic energy (second column), normalized by the mean inlet velocity $u_{in}$, are presented at three stations (see inset; top right panel) along the larynx (A-A'), glottis (B-B') and upper trachea (C-C'), respectively, for two grid resolutions: (i) coarse (9.2M cells) and (ii) fine (25M cells).

this manner, turbulent flow is sustained in the extended pipe section, and a turbulent velocity profile enters the mouth inlet. At the outlet of the model (lower trachea), uniform pressure is prescribed. A no-slip velocity condition is imposed on the airway walls.

The governing equations are discretized using a finite volume method and solved using OpenFOAM, an open-source CFD code [43]. The scheme is second-order accurate in both space and time. To ensure numerical stability the final time step used is $5 \times 10^{-7}$s. A total of 9.2M cells was used to have sufficient grid resolution for LES, based on a grid sensitivity analysis (see S3 Fig). Specifically, normalised time-averaged velocity and turbulent kinetic energy (TKE) predictions for two grid resolutions, namely coarse (9.2M cells) and fine (25M cells), were compared at an inlet Re = 7'000. The comparisons of 1D velocity magnitude profiles at three stations along the larynx (A-A'), glottis (B-B') and upper trachea (C-C') are shown in Fig 2. We find very good agreement between the two grid resolutions, ensuring the adequacy of the selected resolution for LES.

## Results and discussion

### Glottal jet characteristics

Flow through the human larynx, i.e. the passageway connecting the mouth to the respiratory airways, is modulated by a valve-like constriction known as the glottis. During inhalation, this constriction forms a jet of air that extends into the trachea known as the glottal or laryngeal jet. The glottal jet embodies the most dominant flow feature in the upper airways and has been studied extensively in the production of voice and speech [44] as well as in its role in mixing and dissipating boluses of inhaled aerosols [3, 20, 45]. By plotting the velocity magnitude contours for the highest flowrate case (Re = 7'000), we observe the characteristic structure clearly, namely a high velocity flow region extending within and downstream of the constriction and dissipating further downstream (Fig 3a–c). In our idealized geometry, we observe a mostly symmetric glottal jet, while asymmetries would be expected in more realistic patient-specific anatomies; a phenomenon also known as glottal jet skewing [44]. We briefly note that typically, limited velocity data are experimentally resolved in the vicinity of the model's walls (Fig 3a and 3d); this is a well-known limitation of PIV techniques and generally due to the decreasing concentration or loss of tracer particles at the walls, in conjunction with significant flow gradients resulting from the no-slip condition (and possibly wall reflections) [46].

The glottal constriction has been known to represent a source of turbulent flow, despite the relatively low Reynolds numbers ($O(10^3)$) [10]. Shear flows such as the glottal free jet can reduce the critical Reynolds threshold and nevertheless generate turbulent kinetic energy (TKE); a key characteristic associated with the formation of eddies and other coherent flow structures used in classical turbulence analyses. In the examined cases, airflow enters the mouth in the laminar regime at the low flow rate case (Re = 1'500). However, low levels of turbulence develop downstream due to geometrical effects such as bends and constrictions. In contrast, both for the intermediate and high flow rate cases (Re = 4'500 and Re = 7'000), airflow now enters the mouth under turbulent flow conditions. In Fig 3d–f, we plot mean flow TKE contours along the mid-sagittal plane for the three modalities (TPIV, RANS and LES) at the higher Reynolds number case (Re = 7'000). While the velocity magnitude contours agree rather closely between all three modalities, we observe discrepences between TKE results. Firstly, maximum values of TKE differ in the downstream wake of the jet, at the shear interface with the resting fluid; this observation was previously reported for example by Lin et al. [20] in a patient-specific geometry and concurrently by Das et al. using the same identicl idealized mouth-throat geometry [3].

For all three modalities, we observe a common feature with the presence of a thin streak of maximum TKE values originating at the glottic that gradually widens and dissipiates downstream into the trachea. In both experiments and RANS, the shape appears to be in strong agreement whereas for the LES the peak TKE streak is more underlined. However, the RANS simulation reports more than ∼60% lower peak TKE values relative to both TPIV and LES, while a background base level of TKE upstream of the glottis is resolved in the TPIV measurement (i.e. 1–2 m/s) yet absent in RANS (i.e. baseline of near zero TKE) and much lower in LES (i.e. <1 m/s). It is known that differences in TKE hold potential ramifications towards predictions of inhaled aerosols dispersed in the lungs [8]. Notably, the significantly lower TKE levels obtained with RANS are acknowledged to lead to overpredictions of particle deposition when only the (time-averaged) RANS velocity field is used [23, 47]. Indeed, in regions where there is significant large-scale anisotropy in turbulence, turbulent dispersion plays an important role in particle transport and tends to decrease deposition. Hence, RANS are typically used together with a turbulent dispersion model to provide improved deposition predictions. In

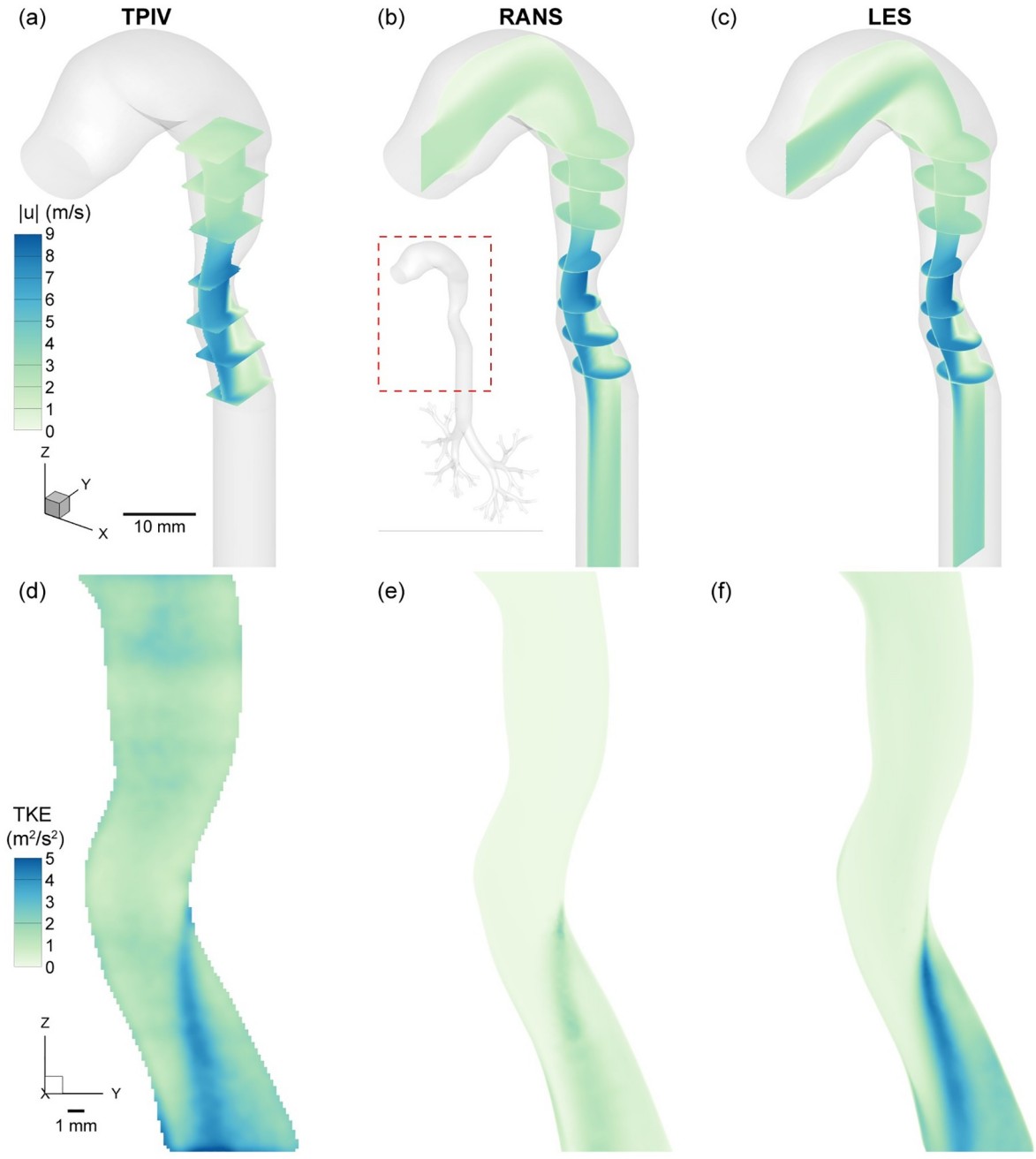

**Fig 3. Comparison of flow characteristics between tomographic PIV (TPIV), Reynolds Averaged Navier Stokes (RANS) and Large Eddy Simulation (LES) for the high flowrate case (Re = 7'000).** Top row: Velocity magnitude contours along the mid-sagittal plane shown along with several orthogonal transverse planes to illustrate 3D characteristics of the flow. Bottom row: Turbulent kinetic energy (TKE) contours are plotted on the mid-sagittal plane. Results are generally in good agreement with strong similarity but derivative TKE values reveal more subtle differences. Specifically, RANS underestimates the peak TKE values near the jet's wake while the TPIV data introduce background noise.

contrast, since LES resolve the large scale eddies, these do not need an additional dispersion model when addressing particle-laden flows [48].

Next, we compare measurements for the low (Re = 1'500) and intermediate (Re = 4'500) flow rate cases, plotted as velocity magnitude contours overlaid with velocity vectors in Fig 4.

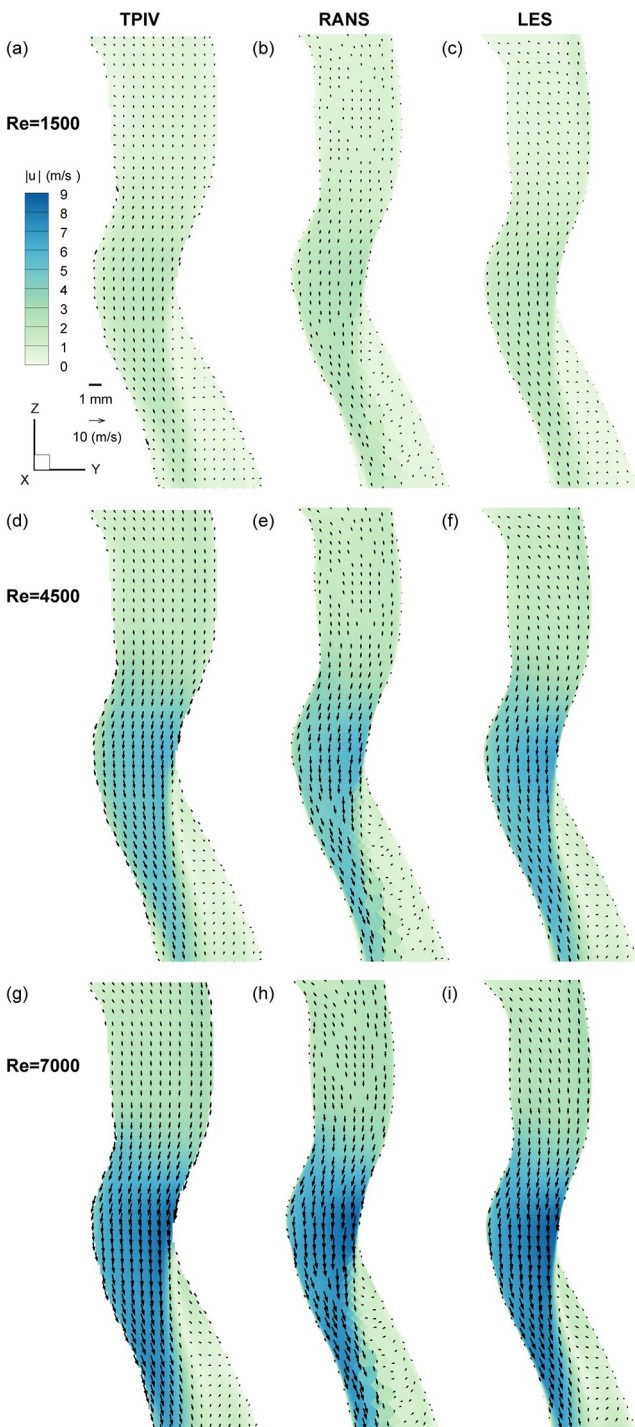

**Fig 4. Flow characterization of the laryngeal jet variation on the mid-sagittal plane in the region of interest (ROI).**
Results are presented as a function of the inlet Reynolds number for TPIV, RANS and LES, respectively.

We briefly note that a small size deviation in the width of the experimental model relative to the numerical ones shown (i.e. a deviation on the order of <5%), likely introduced by the volume calibration step and the physical compliance of the silicone phantom model itself (and also seen in Fig 3d–f). With such differences accounted for, we nevertheless consitently

observe the common laryngeal jet structure described above and present across all flowrates, underlining how the jet phenomenon scales consistently. In the low flowrate case (Re = 1'500) plotted in Fig 4a–c, we observe the weakest jet, with peak velocity values of ∼2 m/s and a ∼1 m/s upstream and downstream flow, with near zero counter flow at the shear interface. For the intermediate flow rate (Re = 4'500) plotted in Fig 4d–f, we observe the same jet structure, with higher peak velocity values (∼5 m/s) and the same near zero counter flow to the right of the shear interface. Lastly, we plot the high flow rate case (Re = 7'000) in Fig 4g–i, with the highest peak velocity values at >9 m/s. We note that the near zero counter flow to the right of the shear interface is nearly identical between the mid and high Re cases, and slightly lower in the low Re case. While the three modalities agree well for the low Re case, we observe in the mid and high Re flows that the RANS simulations does not capture the zero-flow interface as well, which also explains the lower TKE values discussed earlier in Fig 3e, as a common source of energy for turbulent velocity fluctuations lies in the presence of shear in the mean flow.

## Secondary flows

In a next step, we compare ensuing secondary flows downstream of the glottal constriction. Fig 5 plots velocity magnitude contours and velocity vectors for each of the three modalities and flow rates, resulting in a 3 by 3 matrix. Similarly, as seen along the mid-sagittal plane (Fig 4), the basic flow features are largely invariant with respect to Re variation, whereas flow magnitudes vary as anticipated with higher Re number.

Here, we observe a pair of counter-rotating vortices classically refered to as Dean vortices, originating from the curvature of the laryngeal geometry. The Dean number is defined as $Dn = Re\sqrt{D/2r}$, where $r$ denotes the curvature radius and $D$ the cross-sectional diameter of the airway. In our analysis, the Dean number changes only with Re because of the statically defined geometry (i.e., $r$ and $D$ are constants). Therefore our measurements of increasingly stronger Dean vortex flow correlate with increased Re, as plotted in Fig 5. We note here that the cross-section in 2D appears rectangular, in contrast to the circular cross-sections in the numerical simulations (RANS and LES); a consequence of the volume illumination technique which involves knife-edges to form the laser light into a prism (see Fig 1, and subsequent post-processing masking steps that are performed in 2D.

## One-dimensional velocity curves

In a final step, we compare detailed flow characteristics between the experimental and numerical approaches via the simultaneous plotting of 1D velocity magnitude curves. To this end, we first identify the mid-sagittal plane that bisects the geometry (see dashed-dot line in Fig 6a) and the region imaged via TPIV experiments (see Figs 1 and S1). Positioned in this orientation, the mouth inlet is viewed in the normal direction, with a smaller isometric view given as a reference. Four transverse lines (labeled A through D) spanning the mid-sagittal plane are chosen for plotting 1D velocity curves, as shown in Fig 6b.

Good agreement is observed between all three modalities (i.e, TPIV, RANS and LES) for each of the four lines, with variations slightly more pronounced in Line A due to the shorter $y$-axis range relative to Lines B-D. We observe that TPIV and LES velocity magnitude curves along line A are very similar (i.e. LES exceeds the TPIV values by a maximum of 1–2%), characterized by asymmetric twin peaks, with the higher peak on the left side of the dimensionless $x$-axis. The RANS curve, by contrast, features a more symmetric pair of peaks and deviates from the TPIV measurements by <4%. For lines B-D, excellent agreement (<1% deviation) is observed between all modalities.

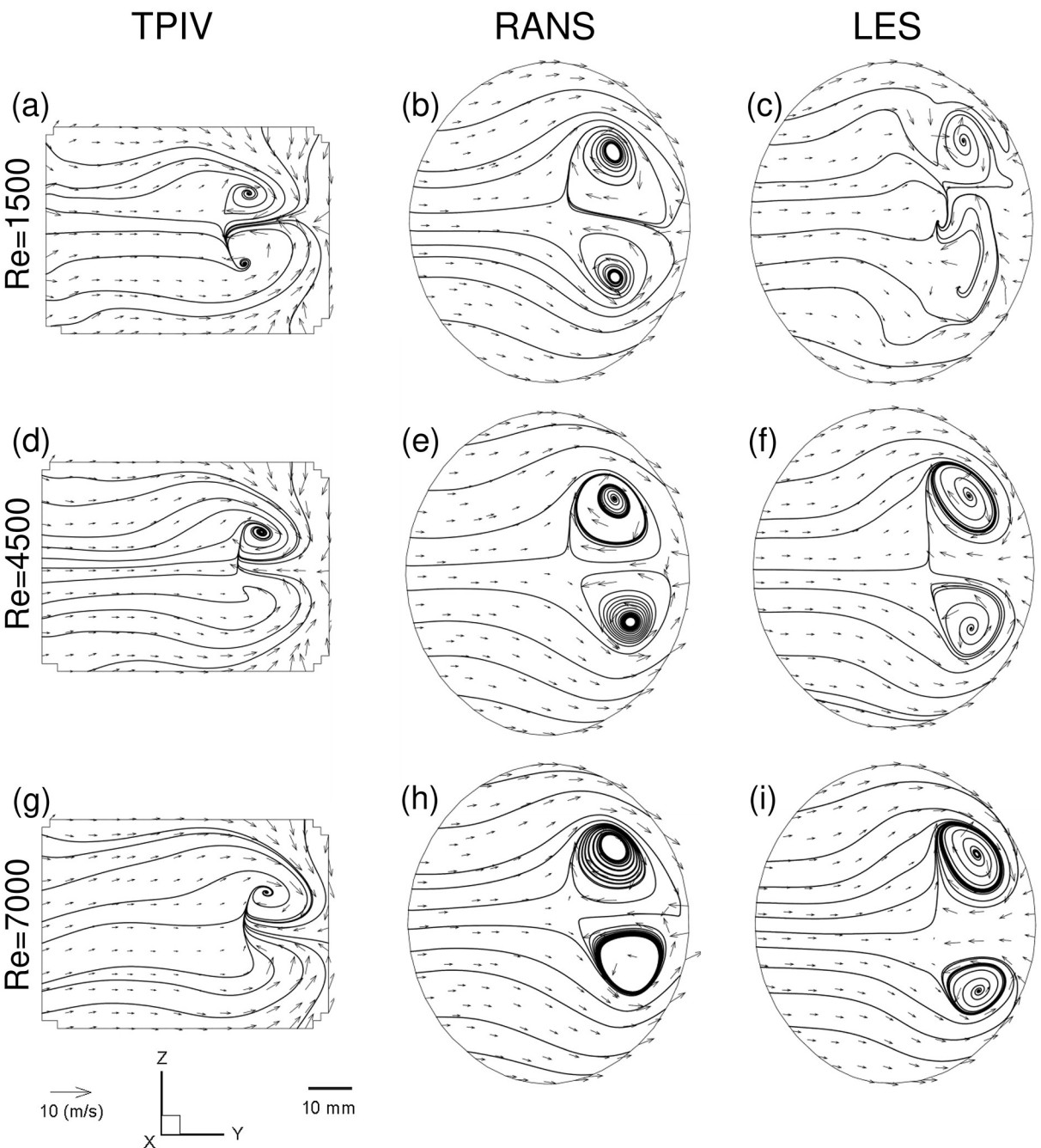

**Fig 5. Flow characterization of the laryngeal jet with Reynolds variation on a transverse plane bisecting the jet's wake.** Results are presented as a function of the inlet Reynolds number for TPIV, RANS and LES, respectively and exemplifies reconstructed flow streamlines in the 2D cut plane (see Line C in Fig 6a for the location of the cut plane).

## Conclusion

The present work has been motivated by the ongoing need for experimentally-resolved 3D flow data to improve the valididty and accuracy of computational fluid dynamics (CFD) simulations resolving turbulent airflows in the upper and extra-thoracic airways towards various

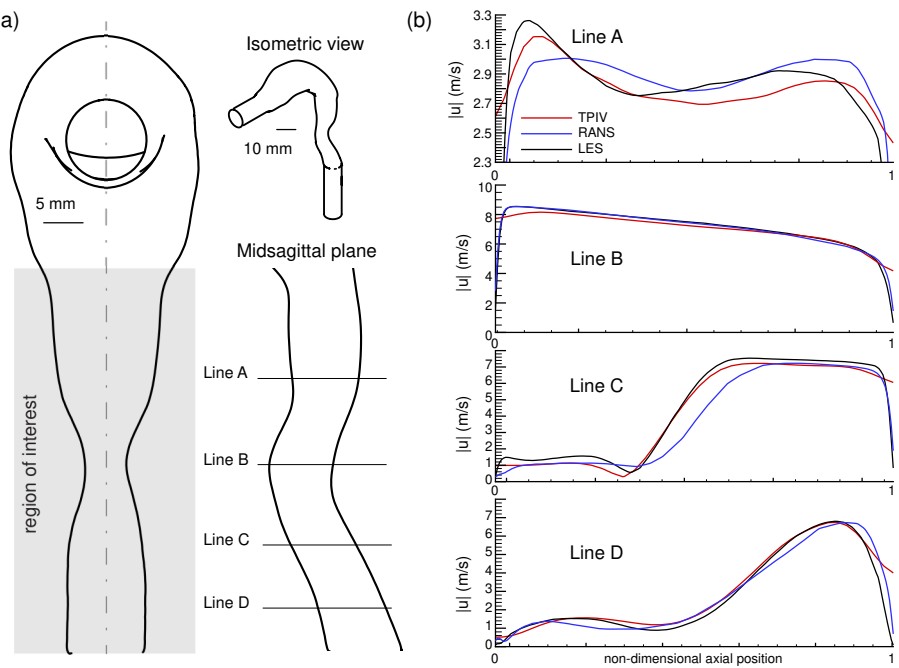

**Fig 6. 1-dimensional velocity magnitude curves for Re = 7'000 case.** (a): Schematic of the full mouth-throat model geometry (see isometric view) used in the fabrication of the experimental phantom and computational mesh. The area illuminated via laser (Region of Interest) and imaged using tomographic particle image velocimetry (TPIV) is noted, along with four lines (labeled A-D) spanning the mid-sagittal plane. (b) Comparison of 1D velocity profiles along the anotated Lines A-D.

inhalation therapy and therapeutic or toxic dosimetry applications. To the best of our knowledge, the findings presented herein are the first detailed 3D *in vitro–in silico* correlations of respiratory airflows in a benchmark anatomical mouth-throat model. We find that a RANS k-$\omega$ SST model adequately predicts velocity flow patterns for Re numbers spanning 1'500 to 7'000, supporting results in close proximity to a more computationally-costly LES model. Yet, RANS significantly underestimates turbulent kinetic energy (TKE), thus demonstrating the superiority of LES as a higher-order turbulence modeling scheme. With a keen eye on end-user applications across various respiratory disciplines, researchers can leverage such validation data in conjunction with open-access files (see SM) for improved predictive CFD models.

## Supporting information

**S1 Fig. Tomographic particle image velocimetry (TPIV) experimental setup.** A: Photographed in the lab B: a schematic illustration.
(EPS)

**S2 Fig. Finite element mesh used in Reynolds-averaged Navier–Stokes (RANS) simulations.**
(TIFF)

**S3 Fig. Large eddy simulation (LES) mesh refinement and model scheme comparison.**
(EPS)

**S1 Text. RANS Equations.**
(PDF)

**S1 Data. Mouth-throat geometry STL file.**
(STL)

**S2 Data. Modified mouth-throat geometry file used in 3D printing the mold for fabricating the silicon phantom.**
(STL)

## Author Contributions

**Conceptualization:** Eliram Nof, Josué Sznitman.

**Data curation:** Eliram Nof, Saurabh Bhardwaj, Pantelis Koullapis.

**Formal analysis:** Saurabh Bhardwaj, Pantelis Koullapis.

**Funding acquisition:** Josué Sznitman.

**Investigation:** Eliram Nof, Saurabh Bhardwaj, Pantelis Koullapis, Josué Sznitman.

**Methodology:** Eliram Nof, Saurabh Bhardwaj, Pantelis Koullapis, Ron Bessler.

**Project administration:** Eliram Nof, Josué Sznitman.

**Resources:** Stavros Kassinos.

**Supervision:** Stavros Kassinos, Josué Sznitman.

**Validation:** Eliram Nof, Saurabh Bhardwaj, Pantelis Koullapis.

**Visualization:** Eliram Nof, Saurabh Bhardwaj, Pantelis Koullapis.

**Writing – original draft:** Eliram Nof, Saurabh Bhardwaj, Pantelis Koullapis, Josué Sznitman.

**Writing – review & editing:** Eliram Nof, Saurabh Bhardwaj, Pantelis Koullapis, Stavros Kassinos, Josué Sznitman.

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
