## [Decision Letter · Decision Letter 0]

7 Nov 2022

Dear Prof. Sznitman,

Thank you very much for submitting your manuscript "In vitro−in silico correlation of three-dimensional turbulent flows in an idealized mouth-throat model" for consideration at PLOS Computational Biology.

As with all papers reviewed by the journal, your manuscript was reviewed by members of the editorial board and by several independent reviewers. In light of the reviews (below this email), we would like to invite the resubmission of a significantly-revised version that takes into account the reviewers' comments.

We cannot make any decision about publication until we have seen the revised manuscript and your response to the reviewers' comments. Your revised manuscript is also likely to be sent to reviewers for further evaluation.

Sincerely,

Alison L. Marsden

Academic Editor

PLOS Computational Biology

Edwin Wang

Section Editor

PLOS Computational Biology

Reviewer's Responses to Questions

**Comments to the Authors:**

Reviewer #1: The study by Nof et al. provides exciting 3D maps of velocity fields from Tomographic Particle Image Velocimetry (TPIV), RANS simulations, and LES simulations of an idealized mouth-throat geometry of a healthy adult. This is the first study to provide 3D TPIV measurements, which can be used to validate computational models. Notably, the authors provide the 3D data through an online repository, which can be used by the community. This is a well written paper and I have just a few minor comments that should be addressed prior to publication.

Thank you for providing the particle relaxation time. Can you provide us with how much a particle is expected to drift during an experiment?

The no-slip wall boundary condition is hard to observe on the cutouts in Fig. 3. Is this because of near wall averaging of the experimental data?

It would be great to include a brief discussion on how the flow profile differences between RANS and LES might lead to differences in particle deposition. Specifically, is it expected that the unresolved TKE might under-estimate deposition?

Reviewer #2: The manuscript describes a comparison of Tomogaphic (3D) PIV measurements with LES and RANS in a mouth throat model. Although the authors report about limitations of already existing studies, their results do not much contribute to new knowledge. It is rather another Benchmark study which shows good agreement between LES and TPIV and larger deviations for the RANS simulations as already reported in their reference [23 - Koullapis et al.]. The advantage of 3D measurements which was emphasized as new in the paper could not be found. The authors showed again detailed comparisons for the mid-sagittal plane and one cross section. The results could have been received also from Stereo PIV (if pure secondary flows had been presented in Fig. 5, 2D planar PIV would have been sufficient to gain the same results). Although additional planes are roughly shown in Fig. 3, their details are not given or discussed. Thus, introduction and conclusion should be changed accordingly.

The following points need to be furthermore addressed:

1. The introduction is too long and should concentrate on the main goal of the paper

2. Important information in the description of the experimental methods are missing. These are

1. The time delay between successive images used for the PIV measurements (for all 3 flow rates)

2. What was the field of view (FOV) in mm and what pixel and vector resolution could have been achieved (see next comment)?

3. The size of the FOV (for each camera) is probably something in the range of 100 x 80 mm^2. With the camera pixels given, one would achieve a resolution of about 80µm/Px. It is mentioned that tracers in the size of 10µm have been used. That means, they do not even cover a pixel. How were the measurements possible and more important, what errors/uncertainties can be expected? Peak locking must have been a big issue. Please elaborate on these discrepancies. This becomes furthermore important when differences to the CFD results are discussed.

4. What was the reason for the use of fluorescent tracers? – as no optical filters have been mentioned.

5. Obviously, the side walls of the mouth throat model were not illuminated anymore but the light was cut off by the knife edges. Please explain why this was done as flow information seems to be missing then for Fig. 5

3. RANS details: the lines 163 – 181 only repeat texbook knowledge. This passage can be omitted

4. Fig 1: The camera lenses are not oriented perpendicular to the phantom model wall. Please elaborate on possible errors due to optical distortions. E.g. Buchmann et al. (Exp Fluids (2011) 50:1131–1151) have used prisms to exclude any optical distortions. Why were these not necessary here?

5. Fig. 3 and 4: The mid-sagittal plane was significantly wider for the TPIV than for CFD. Please elaborate

6. Secondary Flows: In this section Fig. 5 should show secondary flow. However, color coded is again the velocity magnitude. It would be more interesting to visualize only the in plane velocity components to show only the secondary flow. The described dean vortices can be identified only with difficulty.

7. There must be something wrong with the scaling in Fig. 5 for the TPIV. It is mentioned though that only a rectangular cross sectional view could be shown here, but the dimensions do not agree at all with the CFD model. The horizontal extension is much wider than the actual model geometry, the vertical structures seem to be squeezed. i.e. the vertical extension of the low speed region is much smaller for PIV than for the CFD results. At least the characteristic round geometry cannot be recognized for the PIV. This needs correction.

8. In the suppl. Mat. S3 Files, the authors mention a repository for the measured 3D vector field source files. These data could not be found. Please add or correct this.

**Have the authors made all data and (if applicable) computational code underlying the findings in their manuscript fully available?**

Reviewer #1: Yes

Reviewer #2: **No: **

PLOS authors have the option to publish the peer review history of their article (what does this mean?). If published, this will include your full peer review and any attached files.

Reviewer #1: **Yes: **Jessica M Oakes

Reviewer #2: No
---

## [Decision Letter · Decision Letter 1]

8 Feb 2023

Dear Prof. Sznitman,

We are pleased to inform you that your manuscript 'In vitro−in silico correlation of three-dimensional turbulent flows in an idealized mouth-throat model' has been provisionally accepted for publication in PLOS Computational Biology.

Best regards,

Alison L. Marsden

Academic Editor

PLOS Computational Biology

Edwin Wang

Section Editor

PLOS Computational Biology

Please make the one minor correction requested by the reviewer prior to submitting the final version of the paper for publication.

Reviewer's Responses to Questions

**Comments to the Authors:**

Reviewer #2: Almost all issues have been accordingly addressed. Only one discrepancy remains (Lines 112 – 115): It is mentioned that image acquisition was done using a single frame mode with 1250fps and 800us image separation. It the next sentence a pulse time separation of 25 – 40us has been mentioned. This can only apply to double pulse mode, i.e. with double images. As only one of these methods could have been applied, this requires correction.

**Have the authors made all data and (if applicable) computational code underlying the findings in their manuscript fully available?**

Reviewer #2: Yes

PLOS authors have the option to publish the peer review history of their article (what does this mean?). If published, this will include your full peer review and any attached files.

Reviewer #2: No

---

## [Editor Report · Acceptance letter]

16 Mar 2023

PCOMPBIOL-D-22-01317R1 

*In vitro−in silico* correlation of three-dimensional turbulent flows in an idealized mouth-throat model

Dear Dr Sznitman,

I am pleased to inform you that your manuscript has been formally accepted for publication in PLOS Computational Biology. Your manuscript is now with our production department and you will be notified of the publication date in due course.

With kind regards,

Zsofi Zombor
